# Association of hydralazine use with risk of hematologic neoplasms in patients with hypertension: A nationwide population-based cohort study in Taiwan

Li-Tzu Wang[1,2◉], Wu-Chien Chien[3,4,5◉], Kevin Sheng-Kai Ma[6], Chi-Hsiang Chung[3,4], Yeu-Chin Chen[7], Wei-Che Tsai[8], Bing-Heng Yang[9,10]*

1 School of Medical Laboratory Science and Biotechnology, College of Medical Science and Technology, Taipei Medical University, Taipei, Taiwan, 2 Ph.D. Program in Medical Biotechnology, College of Medical Science and Technology, Taipei Medical University, Taipei, Taiwan, 3 Graduate Institute of Public Health, College of Public Health, National Defense Medical University, Taipei, Taiwan, 4 Department of Medical Research, Tri-Service General Hospital, National Defense Medical University, Taipei, Taiwan, 5 Taiwanese Injury Prevention and Safety Promotion Association, Taipei, Taiwan, 6 Center for Global Health, Perelman School of Medicine, University of Pennsylvania, Philadelphia, United States of America, 7 Division of Hematology and Oncology, Department of Internal Medicine, Tri-Service General Hospital, National Defense Medical University, Taipei, Taiwan, 8 Division of Cardiology, Department of Internal Medicine, Tri-Service General Hospital, National Defense Medical University, Taipei, Taiwan, 9 Division of Clinical Pathology, Department of Pathology, Tri-Service General Hospital, National Defense Medical University, Taipei, Taiwan, 10 Graduate Institute of Pathology and Parasitology, College of Medicine, National Defense Medical University, Taipei, Taiwan

◉ These authors contribute equally as co-first authors to this work.

* rodancer0629@gmail.com

## Abstract

### Background

Onco-hypertension recognizes well-controlled blood pressure as a favorable prognostic factor for survival in patients with hypertension and solid tumors, including hematologic neoplasms. However, it remains unknown whether continuous use of hydralazine—an antihypertensive agent (AHA) with notable anti-neoplastic activity—is associated with a lower risk of hematologic neoplasms compared to other AHAs.

### Method and findings

Utilizing Taiwan's National Health Insurance Research Database, we conducted a 16-year follow-up study (2000–2015) involving 375,107 patients with hypertension treated with an AHA for ≥180 days. The patients with hypertension were divided into two groups based on hydralazine prescription duration: an exposure group (hydralazine ≥180 days; $n = 59,786$) and a reference group (hydralazine <180 days; $n = 239,144$) after 1:4 matching for sex, age, and index date with the exposure group. Both groups were well-matched, with a mean age of approximately 60.8 years and 52.19% male. We assess the association between hydralazine use and the risk of

which permits unrestricted use, distribution, and reproduction in any medium, provided the original author and source are credited.

**Data availability statement:** The data sets used in this study are held by the Taiwan Ministry of Health and Welfare (MOHW). Requests for access to the data sets must be approved by the MOHW. The data sets can be requested by any researcher interested in accessing them. Please visit the website of the National Health Informatics Project of the MOHW (https://dep.mohw.gov.tw/dos/np-2497-113.html) or contact the National Health Insurance Database (nhird@nhri.edu.tw). Address of Taiwan Ministry of Health and Welfare: No. 488, Sec. 6, Zhongxiao E. Rd., Nangang Dist., Taipei City 115204, Taiwan (R.O.C.) Tel: (+886)2-8590-6666. Fax: (+886)2-8590-6000.

**Funding:** This work was supported by the Tri-Service General Hospital Foundation (TSGH-B-112018 to BHY; TSGH-E-112226 to BHY; TSGH-B-113026 to BHY; TSGH-E-113248 to BHY; TSGH-B-114025 to BHY; TSGH-E-114247 to BHY; TSGH-B-112020 to WCC; TSGH-B-113025 to WCC; TSGH-B-114022 to WCC). The funder's website is: www.tsgh.ndmutsgh.edu.tw/english. This work was also supported by the National Science and Technology Council, Taiwan (NSTC 114-2314-B-038-089-MY3 to LTW). The funder's website is: www.nstc.gov.tw/?l=en). The funders had no role in study design, data collection and analysis, decision to publish, or preparation of the manuscript.

**Competing interests:** The authors have declared that no competing interests exist.

**Abbreviations:** adjusted HRs, adjusted hazard ratios; adjusted sHR, adjusted subdistribution hazard ratio; AHA, antihypertensive agent; AML, acute myeloid leukemia; CCI_R, Charlson Comorbidity Index_revised; CHF, congestive heart failure; CVD, cardiovascular disease; DDDs, defined daily doses; DNMT, DNA methyltransferase; HAAEs, hydralazine-associated lupus-like adverse effects; ICD-9-CM, International Classification of Diseases, 9th Revision, Clinical Modification; IHD, ischemic heart disease; LGTD, Longitudinal Generation Tracking Database; MDS, myelodysplastic syndrome; MM, multiple myeloma; NHIRD, Taiwan's National Health Insurance Research Database; NTD, New Taiwan Dollars; STROBE, Strengthening the Reporting of Observational Studies in Epidemiology.

hematologic neoplasms using Kaplan–Meier analysis and multivariable Cox proportional hazards regression, with models adjusted for concomitant medications possessing potential anti-neoplastic properties. The 16-year cumulative incidence of hematologic neoplasms was lower in the exposure group (105.58 per 100,000 person-years) than in the reference group (160.33). Accounting for death as competing risk, the exposure group exhibited an adjusted subdistribution hazard ratio (adjusted sHR) of 0.789 (95% confidence interval [0.667,0.913]; $P<.001$) for hematologic neoplasms compared to the reference group. Subgroup analyses demonstrated that the association with a lower risk was strongest in the longest prescription duration category. For example, for patients with prescription durations of ≥668 days, the adjusted sHR was 0.448 (95% CI [0.366,0.555]; $P<.001$) for other malignant neoplasms of lymphoid and histiocytic tissue, 0.552 (95% CI [0.453,0.683]; $P<.001$) for multiple myeloma and immunoproliferative neoplasms, and 0.555 (95% CI [0.457,0.689]; $P<.001$) for myeloid leukemia. The main limitation was the potential for residual confounding due to the unavailability of lifestyle and laboratory data in the administrative database.

## Conclusions

In this study, we observed that long-term hydralazine use in patients with hypertension was associated with a lower, duration-dependent risk of hematologic neoplasms. These findings warrant prospective studies to confirm this association and its potential clinical implications.

### Author summary

#### Why was this study done?

- High blood pressure is recognized as a risk factor associated with the development of blood cancers.

- Laboratory studies have shown that hydralazine has biological activities that could counter the mechanisms of blood cancers, such as inhibiting an enzyme called DNA methyltransferase.

- Despite these promising laboratory findings, there was a significant knowledge gap, as no large-scale, population-based study had investigated whether taking hydralazine long-term was actually associated with a lower risk of these cancers in people.

#### What did the researchers do and find?

-  We used a national health database from Taiwan to analyze the health records of nearly 300,000 people with high blood pressure over a 16-year period.

- We compared a group of 59,786 patients who took the drug hydralazine for at least 180 days to a group of 239,144 patients who took other common blood pressure medications.

- After adjusting for other health factors, we found that the group taking hydralazine long-term had an approximately 21% lower risk of being diagnosed with a blood cancer.

### What do these findings mean?

- Our results suggest a link between the long-term use of hydralazine and a lower risk of developing blood cancers in this population of patients with hypertension.

- Because this study only observed patients over time and could not account for lifestyle factors or how well patients took their medication, our findings do not prove that hydralazine causes the lower risk.

- These results highlight the need for future research to confirm the association and understand what it could mean for treating patients with high blood pressure who are at a higher risk for blood cancers.

## Introduction

Onco-hypertension [1] is an emerging field that recognizes well-controlled blood pressure as a favorable prognostic factor for survival in patients with hypertension and solid tumors or hematologic neoplasms such as high-grade hematological malignancies (HMs) [2]. The link between hypertension and hematologic neoplasms is incompletely understood, especially in high-risk settings, such as after allogeneic hematopoietic cell transplantation, where endothelial injury is a critical pathogenic mechanism driving hypertension [3]. Furthermore, the use of novel targeted agents, including tyrosine kinase inhibitors, can lead to drug-induced hypertension [4], further complicating patient management. This challenge underscores the importance of optimal antihypertensive agents (AHAs) selection, as several agents have not only been associated with a lower risk of specific hematologic neoplasm [5] but also show potential mechanistic action, such as direct anti-proliferative effects [6] or inhibition of pro-tumorigenic β-adrenergic signaling [7]. Moreover, growing bodies of evidence indicate that AHAs can be repurposed for the treatment of hematologic neoplasms by targeting specific biological mechanisms [8–10]. An association between a specific AHA and a lower risk of hematologic neoplasms, relative to other AHAs, would suggest that the drug possesses pleiotropic effects beyond its primary function of blood pressure regulation.

Targeting host susceptibility factors has emerged as a promising strategy for reducing the risk of HMs, especially in high-risk populations [11,12]. This approach is motivated by evidence linking HM development to specific driver genes, such as *DNMT3A* [13–15] and *TET2* [13–15], as well as independent risk factors like hypertension [2,4] and hepatitis B virus infection [2,16]. Accordingly, an ideal agent for reducing the risk of hematologic neoplasms would target multiple pathogenic pathways, including both driver genes and independent risk factors. This highlights a critical gap in the literature: while AHAs possess diverse mechanisms of action and are widely used, their association with the overall risk of hematologic neoplams remains insufficiently investigated.

Certain AHAs have been repurposed for the treatment of specific hematologic neoplasms [8–10]. Hydralazine—an arterial vasodilator and a DNA methyltransferase (DNMT) inhibitor—is among these AHAs and has been repurposed for the treatment of T-cell leukemia [17], cutaneous T-cell lymphoma [18], and myelodysplastic syndrome (MDS) [18,19]. Hydralazine has also been reported to suppress DNMT3a expression [20–22], and *DNMT3A* mutations are implicated in the development of various hematologic neoplasms [23], including myeloproliferative neoplasms, MDS [24], acute myeloid leukemia (AML) [25–27], and T-cell lymphoma [28]. Furthermore, hydralazine was observed to increase TP53 activity [29], a factor involved in de novo AML [26,27] and lymphomagenesis [30]. Owing to its unique pharmacological profile among AHAs, hydralazine warrants investigation for its potential association with the risk of hematologic neoplasms in patients with hypertension.

According to our review of the literature, the association between AHA use and hematologic neoplasm risk has yet to be evaluated by a large-scale population-based study. Accordingly, to fill this research gap, we used a nationwide database to analyze hematologic neoplasm risk in patients with hypertension receiving hydralazine versus other AHAs.

## Methods

### Ethics statement

The study protocol was approved by the Institutional Review Board of Tri-Service General Hospital (TSGHIRB No. E202216031). The board waived the requirement for informed consent because of the anonymization of all extracted data.

### Data source

This retrospective cohort study employed data extracted from the Longitudinal Generation Tracking Database (LGTD) 2000–2015. The LGTD is a subset of Taiwan's National Health Insurance Research Database (NHIRD) and encompasses the health records of 1,936,512 patients [31]. Taiwan's National Health Insurance program provides coverage for ≥99.9% of the country's 23 million residents [31–33], with the NHIRD serving as the claims database for this program. From the LGTD, we extracted information on the patients' clinicodemographic characteristics (such as age, sex, and residence area), diagnoses, treatments, and surgical history. Diagnoses were coded using *International Classification of Diseases, 9th Revision, Clinical Modification* (*ICD-9-CM*) diagnostic codes. Notably, in contrast to unvalidated *ICD-10-CM* diagnostic codes, the *ICD-9-CM* diagnostic codes in the NHIRD have been validated to have high sensitivity for hypertension (92.4%) [34] and all cancers (91.5%) [35]. Prior to data extraction, all confidential information, such as medical institutes and patient names, were encrypted to ensure privacy.

### Study population and AHA treatments

The study population was selected from LGTD 2000–2015. As a preliminary step to confirm the association between hypertension and hematologic neoplasm in our population, we first identified a cohort of patients with hypertension (*ICD-9-CM* codes 401–405). This cohort was then matched using propensity scores to patients without hypertension at a 1:4 ratio based on age, sex, and index date (Fig 1). This step confirmed that hypertension was an independent risk factor for hematologic neoplasms in our cohort, providing the rationale for the primary analysis. Medication exposure was standardized by converting all prescription doses into defined daily doses (DDDs), as specified by the World Health Organization Collaborating Centre for Drug Statistics Methodology. The DDD is the assumed average maintenance dose per day for a drug used for its primary indication in adults. For hydralazine, the DDD is 0.1 g per day. The cumulative exposure for each patient was calculated by dividing the total prescribed dose of hydralazine recorded in the database by its DDD. This method allowed for a standardized assessment of exposure duration and for classifying patients into an exposure group (≥180 days of cumulative use) and a reference group (<180 days of cumulative use). To enhance comparability, we randomly selected a subset of patients from the reference group for propensity score matching with those in the exposure group in terms of age, sex, and index date at a 4:1 ratio. Considering that the diagnostic codes for hypertension may not have been recorded for some patients with hypertension receiving regular AHA treatment, which would have resulted in an underestimation of the study population, we also included AHA-treated patients who had received a hypertension diagnosis within the 2 years prior to the index date. The cumulative incidence of hematologic neoplasm was estimated using Kaplan–Meier curves. We employed the 2020 International Society of Hypertension global hypertension practice guidelines [36–38] for the selection of AHAs, namely the A/C/D classes of AHAs: A (angiotensin-converting enzyme inhibitors or angiotensin receptor blockers: quinapril hydrochloride, lisinopril, fosinopril sodium, enalapril maleate, perindopril, captopril, benazepril hydrochloride, and ramipril), C (calcium channel blockers: nifedipine, felodipine, nicardipine, amlodipine besylate, verapamil hydrochloride, and diltiazem hydrochloride), and D

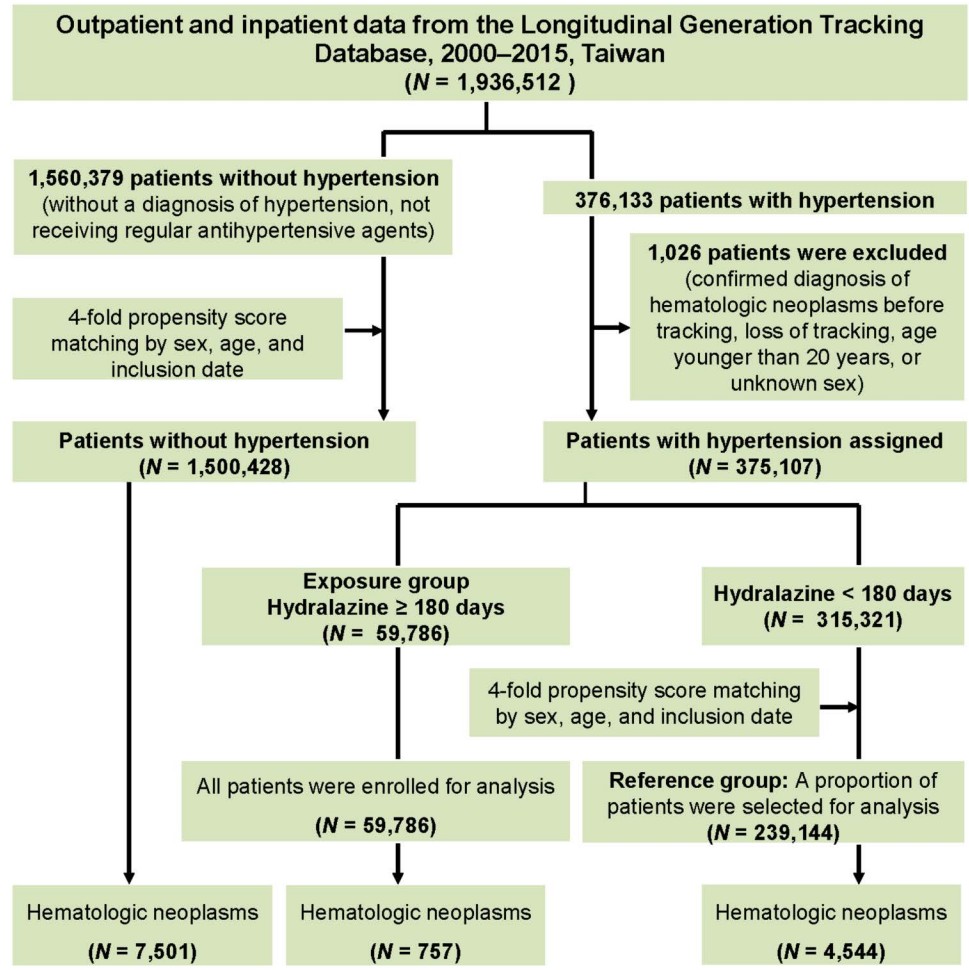

**Fig 1. Population-based analysis of hematologic neoplasm development in patients without hypertension and patients with hypertension receiving regular antihypertensive agents.**

(thiazide-like diuretics: chlorthalidone, chlorothiazide, indapamide, and metolazone), along with hydralazine. Other AHAs (spironolactone, α-blockers, and β-blockers) were only considered for specific indications (e.g., hyperkalemia, atrial fibrillation, heart failure, angina, and younger women who were pregnant or were planning pregnancy) [36] not for general use, and thus we did not include them in this study. Only patients who received any (single or combination therapy) of the aforementioned A/C/D classes of AHAs and hydralazine successively for ≥180 days were included. After confirming patient eligibility, we calculated person-time for exposed and unexposed patients. For exposed patients, person-time was calculated from the date they accrued ≥180 days of hydralazine exposure. For unexposed patients, person-time was calculated from the date they accrued ≥180 days of exposure to A, C, or D classes of AHAs. Person-time was measured from the start of follow-up until the date of hematologic neoplasm development, date of death, or the end of follow-up, whichever occurred first. Regarding the exclusion criteria, participants who were aged <20 years, were lost to follow-up, received a diagnosis of hematologic neoplasm before the index date, or had missing demographic information excluded. To assess potential selection bias, we compared baseline demographic and clinical characteristics between the included patients (exposure group, $n = 59,786$) and those initially excluded from the population of patients with hypertension ($n = 1,026$) (Table A in S1 File).

## Covariates and comorbidities

We employed sex, age (20–29, 30–39, 40–49, 50–59, or ≥60 years), season of index date, residence area, urbanization level (1: ≥1,250,000 people; 2: 500,000–1,249,999 people; 3: 150,000–499,999 people; or 4: <149,999 people), health insurance premiums, and hospital level (medical center, regional hospital, or local hospital) as covariates. Health insurance premiums, denominated in New Taiwan Dollars (NTD), are calculated based on income levels and serve as a reliable proxy for the patient's socioeconomic status within Taiwan's National Health Insurance system. In this study, premiums were categorized into three groups as NTD per month: <18,000, 18,000–34,999, and ≥35,000 (1 NTD = ~0.03 USD).

We also adjusted potential confounders, such as the comorbidities [39] or concomitant medications [8,40] in which previous studies have reported direct or indirect associations with hematologic neoplasm development (Table B in S1 File). Additionally, the Charlson Comorbidity Index_revised (CCI_R) was used to evaluate the overall extent of the comorbidity-associated hematologic neoplasm risk.

## Outcome measure

The primary outcome was the occurrence of any hematologic neoplasm event in a patient. A hematologic neoplasm event was identified on the basis of the presence of any of the following *ICD-9-CM* codes: (1) 200 (lymphosarcoma and reticulosarcoma); (2) 201 (Hodgkin's disease); (3) 202 (other malignant neoplasms of lymphoid and histiocytic tissue); (4) 203 [multiple myeloma (MM) and immunoproliferative neoplasms]; (5) 204 (lymphoid leukemia); (6) 205 (myeloid leukemia); (7) 206 (monocytic leukemia); (8) 207 (other specified leukemia); (9) 208 (leukemia of unspecified cell type); (10) 238.4, 238.5, 238.6, 238.71–238.76, 238.79, or 289.83 (neoplasm of uncertain behavior); (11) 238.72–238.75 (MDS); (12) 273.1–273.3 or 273.8–273.9 (paraproteinemia); and (13) 289.0 or 289.6 (other polycythemia; Table B in S1 File).

The onset and long-term progression of hematologic neoplasms were analyzed using two models (Table C in S1 File). The first (first-event) model was based on initial diagnosis to assess the risk of developing hematologic neoplasm for the first time. Given that the clinical course of hematologic neoplasms over 16 years can be complex, the second (multiple-event) model included all hematologic neoplasm events from each patient to evaluate cumulative disease burden. The use of both models enabled us to conduct a comprehensive analysis: the first-event model could capture the initial risk of hematologic neoplasm, whereas the multiple-event model could capture the cumulative burden and progression of the disease, thereby enhancing the understanding of both hematologic neoplasm onset and long-term outcomes. Because patients could receive multiple hematologic neoplasm diagnoses during the follow-up period, an overall adjusted subdistribution hazard ratio (adjusted sHR) could not be calculated in the multiple-event model. The association between the incidence of hematologic neoplasms and mortality in the exposure group was evaluated by calculating hematologic neoplasm-related and all-cause mortality. Participants with any diagnosis of hematologic neoplasm on the date of their mortality were considered as having hematologic neoplasm-related mortality.

## Statistical analysis

Intergroup comparisons of categorical variables were performed using a chi-squared test or Fisher's exact test, depending on whether the proportion of all categorical outcomes was >5% or any proportion was <5%, respectively. Continuous variables were compared using a *t* test or one-way analysis of variance with Scheffe's post hoc test. To assess the cumulative incidence of hematologic neoplasms, the log-rank test was employed, and the results were visualized using Kaplan–Meier curves. Associations with hematologic neoplasms were determined using multivariable Cox regression analyses, with results presented as adjusted hazard ratios (adjusted HRs) along with 95% confidence intervals. Statistical significance was set at a two-tailed *P* value of <.001. This stringent threshold was chosen to minimize the risk of false positives (type I error) owing to the large sample size and multiple statistical comparisons. Schoenfeld's global test was conducted using STATA 9.0 to evaluate the proportionality assumption of covariates and comorbidities [41]. To account for the potential impact of disproportionate subgroup distributions on the overall results, we conducted a leave-one-out analysis by excluding cases from any specific subgroup that constituted >30% of the study population. Moreover, a sensitivity analysis was performed by excluding patients who

received a hematologic neoplasm diagnosis within the first few years of tracking. For estimating the competing risk of mortality, Fine and Gray's competing risk model was constructed, with all-cause mortality serving as a covariate [42]. The hematologic neoplasm risk was estimated using two models: adjusted HRs (based on Cox regression), and adjusted sHRs (based on Fine and Gray's competing risk models) (Table 2); the corresponding unadjusted (crude) hazard ratios are provided in the Table K–TableQ in S1 File. All analyses were conducted using SPSS (version 22.0; IBM, Armonk NY, USA). This study followed the Strengthening the Reporting of Observational Studies in Epidemiology (STROBE) guidelines (S1 Checklist).

## Results

### Patient selection and characterization

The study population was established through two 1:4 matching procedures (Fig 1). First, we identified a primary cohort of 375,107 patients with hypertension and matched them to 1,500,428 patients without hypertension (selected from a pool of 1,560,379 individuals). Second, within the cohort of patients with hypertension, we defined an exposure group ($n = 59,786$) and a reference pool ($n = 315,321$). We then matched the exposure group 1:4 to this reference pool, yielding a final reference group of 239,144. The 16-year cumulative hematologic neoplasm incidence in the exposure group was significantly lower than that in the reference group (105.58 versus 160.33 per 100,000 person-years).

To check for selection bias, we compared baseline demographic and clinical characteristics between the exposure group ($n = 59,786$) and excluded patients ($n = 1,026$) (Table A in S1 File). The excluded individuals were significantly younger (mean age 52.18 versus 60.82 years, $P < .001$) and had a significantly higher proportion of men (66.08% versus 52.19%, $P < .001$); in addition, the excluded individuals had a significantly higher comorbidity burden, evidenced by higher rates of CHF (5.46% versus 1.33%, $P < .001$) and higher mean CCI_R scores (1.05 versus 0.82, $P < .001$).

Because of the propensity score matching process, age (60.79 versus 60.82 years in the reference group and exposure group), sex (male-to-female ratio: approximately 1.09 in both), and index date (proportions in the 4 seasons) were comparable between the 2 groups. More than half of the patients in both groups were older than 60 years. Regarding residence area, approximately one-third of the patients resided in northern Taiwan, and approximately 70% of them resided in high urbanization levels of the city (1 and 2). The exposure group had a significantly higher comorbidity burden, with higher rates of congestive heart failure (CHF), ischemic heart disease (IHD), malignant neoplasm of the kidney/renal pelvis, and acute glomerulonephritis/nephrotic syndrome, as well as a higher overall CCI_R score. Furthermore, a larger proportion of patients in the exposure group received aspirin, celecoxib, thalidomide, valproate, auranofin, ivermectin, curcumin, or axitinib (all $P < .001$), while a smaller proportion received mebendazole ($P < .001$) and statins ($P = .027$) (Table 1).

### Risk factors associated with hematologic neoplasms

After adjustment of potential confounders, hypertension was associated with a higher risk of hematologic neoplasms (adjusted sHR = 1.483, 95% confidence interval [1.397,1.654]; $P < .001$; Table D in S1 File). Patients with IHD, vascular insufficiency of the intestine, hepatitis B virus infection with or without hepatic coma, malignant neoplasm of the kidney, acute glomerulonephritis or nephrotic syndrome, or a higher CCI_R score were at higher risk of hematologic neoplasms (Table 2). We also examined the associations for other medications known to have anti-hematological neoplasm properties. Mebendazole, which has recognized antileukemia activity [43], was associated with a lower risk of hematologic neoplasms (adjusted sHR = 0.696, 95% CI [0.286,0.957]; $P = .003$), although this association did not meet our prespecified significance threshold. Similarly, no statistically significant associations were observed for other medications investigated: itraconazole [44] and metformin [45] were linked to a lower risk, whereas valproate [46] was linked to a higher risk. However, a higher risk of hematologic neoplasms was associated with the use of aspirin, celecoxib, statins, and thalidomide—with the associations for celecoxib (adjusted sHR = 1.608, 95% CI [1.163,1.933]; $P < .001$) and thalidomide (adjusted sHR = 1.846, 95% CI [1.194,2.466]; $P < .001$) meeting our pre-specified significance threshold—all of which were previously documented to play a therapeutic role in various types of hematologic neoplasms [47].

**Table 1. Baseline characteristics of patients with hypertension by prescription duration of hydralazine, 2000–2015.**

| Variables | Hydralazine | <180 days | | ≥180 days | | P* |
|---|---|---|---|---|---|---|
| | | *n* | % | *n* | % | |
| **Total** | | 239,144 | | 59,786 | | |
| **Sex** | | | | | | Matched |
| Male | | 124,800 | 52.19 | 31,200 | 52.19 | |
| Female | | 114,344 | 47.81 | 28,586 | 47.81 | |
| **Age (years)** | | 60.79±13.82 | | 60.82±13.86 | | Matched |
| **Age group (years)** | | | | | | Matched |
| 20–29 | | 1,764 | 0.74 | 441 | 0.74 | |
| 30–39 | | 14,064 | 5.88 | 3,516 | 5.88 | |
| 40–49 | | 41,832 | 17.49 | 10,458 | 17.49 | |
| 50–59 | | 44,260 | 18.51 | 11,065 | 18.51 | |
| ≥60 | | 137,224 | 57.38 | 34,306 | 57.38 | |
| **Insured premium (NTD)** | | | | | | <.001 |
| <18,000 | | 209,465 | 87.59 | 52,341 | 87.55 | |
| 18,000–34,999 | | 19,127 | 8.00 | 4,778 | 7.99 | |
| ≥35,000 | | 10,552 | 4.41 | 2,667 | 4.46 | |
| **Normal pregnancy** | | 30,597 | 12.79 | 6,475 | 10.83 | <.001 |
| **Comorbidities** | | | | | | |
| CHF | | 978 | 0.41 | 798 | 1.33 | <.001 |
| PE | | 174 | 0.07 | 33 | 0.06 | .895 |
| GI hemorrhage | | 466 | 0.19 | 120 | 0.20 | .784 |
| Cerebral thrombosis | | 370 | 0.15 | 145 | 0.24 | .001 |
| IHD | | 2,570 | 1.07 | 1,014 | 1.70 | <.001 |
| Vascular insufficiency of intestine | | 682 | 0.29 | 198 | 0.33 | .874 |
| Obesity | | 227 | 0.09 | 67 | 0.11 | .711 |
| Malignant neoplasm of kidney/renal pelvis | | 5,701 | 2.38 | 1,978 | 3.31 | <.001 |
| Acute glomerulonephritis/Nephrotic syndrome | | 1,235 | 0.52 | 484 | 0.81 | <.001 |
| Proteinuria | | 1,040 | 0.43 | 333 | 0.56 | .044 |
| Gestational hypertension | | 1,885 | 0.79 | 482 | 0.81 | .385 |
| Asthma | | 16,451 | 6.88 | 3,327 | 5.56 | .002 |
| CCI_R | | 0.78±1.09 | | 0.82±1.22 | | <.001 |
| **Medications** | | | | | | |
| Aspirin | | 33,240 | 13.90 | 8,976 | 15.01 | <.001 |
| Celecoxib | | 27,015 | 11.30 | 7,378 | 12.34 | <.001 |
| Itraconazole | | 12,024 | 5.03 | 2,885 | 4.83 | .152 |
| Mebendazole | | 33,978 | 14.21 | 8,125 | 13.59 | <.001 |
| Leflunomide | | 16,625 | 6.95 | 3,876 | 6.48 | .208 |
| Thalidomide | | 23,154 | 9.68 | 6,022 | 10.07 | <.001 |
| Valproate | | 18,784 | 7.85 | 5,227 | 8.74 | <.001 |
| Metformin | | 38,887 | 16.26 | 9,896 | 16.55 | .345 |
| Auranofin | | 10,245 | 4.28 | 3,542 | 5.92 | <.001 |
| Statins | | 32,973 | 13.79 | 7,896 | 13.21 | .027 |
| Bisphosphonates | | 21,879 | 9.15 | 5,014 | 8.39 | .001 |
| Bromocriptine | | 23,151 | 9.68 | 6,156 | 10.30 | .264 |

*(Continued)*

**Table 1.** (Continued)

| Variables | Hydralazine | <180 days | | ≥180 days | | P* |
|---|---|---|---|---|---|---|
| | | n | % | n | % | |
| Chlorprothixene | | 27,774 | 11.61 | 7,013 | 11.73 | .396 |
| Clotrimazole | | 22,086 | 9.24 | 5,882 | 9.84 | .452 |
| Quinacrine | | 20,274 | 8.48 | 4,782 | 8.00 | .771 |
| Ivermectin | | 17,425 | 7.29 | 5,079 | 8.50 | <.001 |
| Verteporfin | | 18,834 | 7.88 | 3,846 | 6.43 | <.001 |
| Clarithromycin | | 9,795 | 4.10 | 2,115 | 3.54 | .567 |
| Hydroxychloroquine | | 23,401 | 9.79 | 5,357 | 8.96 | .488 |
| Tofacitinib | | 22,673 | 9.48 | 6,014 | 10.06 | .004 |
| Gefitinib | | 24,852 | 10.39 | 5,511 | 9.22 | .006 |
| Curcumin | | 10,565 | 4.42 | 4,056 | 6.78 | <.001 |
| Chlorhexidine | | 12,098 | 5.06 | 3,798 | 6.35 | .278 |
| Axitinib | | 8,920 | 3.73 | 2,458 | 4.11 | <.001 |
| **Season of index date** | | | | | | Matched |
| Spring (Mar–May) | | 59,592 | 24.92 | 14,898 | 24.92 | |
| Summer (Jun–Aug) | | 60,828 | 25.44 | 15,207 | 25.44 | |
| Autumn (Sep–Nov) | | 55,128 | 23.05 | 13,782 | 23.05 | |
| Winter (Dec–Feb) | | 63,596 | 26.59 | 15,889 | 26.59 | |
| **Location** | | | | | | <.001 |
| Northern Taiwan | | 90,023 | 37.64 | 22,518 | 37.66 | |
| Middle Taiwan | | 72,251 | 30.21 | 17,184 | 28.74 | |
| Southern Taiwan | | 42,279 | 17.68 | 11,297 | 18.90 | |
| Eastern Taiwan | | 30,201 | 12.63 | 7,022 | 11.75 | |
| Outlets islands | | 4,390 | 1.84 | 1,765 | 2.95 | |
| **Urbanization level** | | | | | | <.001 |
| 1 (The highest) | | 89,876 | 37.58 | 21,449 | 35.88 | |
| 2 | | 77,245 | 32.30 | 19,780 | 33.08 | |
| 3 | | 30,121 | 12.60 | 8,245 | 13.79 | |
| 4 (The lowest) | | 41,902 | 17.52 | 10,312 | 17.25 | |
| **Levels of hospitals** | | | | | | <.001 |
| Medical center | | 83,972 | 35.1 | 20,745 | 34.70 | |
| Regional hospital | | 82,121 | 34.34 | 20,110 | 33.64 | |
| Local hospital | | 73,051 | 30.55 | 18,931 | 31.66 | |

*P: Chi-squared test was used for all categorical variables, whereas the t test was used for continuous variables.

NTD, New Taiwan dollar; CHF, congestive heart failure; PE, pulmonary embolism; GI, gastrointestinal; IHD, ischemic heart disease; CCI_R, Charlson Comorbidity Index_Revised.

## Association of hematologic neoplasm incidence stratified by hematologic neoplasm subgroup and duration of hydralazine prescription

After accounting for the competing risk of mortality, we observed a duration-dependent association between hydralazine use and a lower risk of overall hematologic neoplasms (Table 3). In the exposure group, patients were categorized into three sub-groups based on prescription duration: 180–350 days, 351–667 days, and ≥668 days. As detailed in Table 3, the adjusted HRs were 0.884 (95% confidence interval [0.743,1.098]; P=.189) for the 180–350 days subgroup, 0.728 (95% CI [0.598,0.904]; P<.001) for the 351–667 days subgroup, and 0.646 (95% CI [0.531,0.803]; P<.001) for the ≥668 days subgroup.

 

**Table 2. Multivariable risk regression analysis of hematologic neoplasm development in patients with hypertension in competing risk model[*].**

| Variables | No competing risk model | | | | Fine and Gray's competing risk model[†] | | | |
|---|---|---|---|---|---|---|---|---|
| | Adjusted HR[‡] | 95% CI | | P | Adjusted sHR[§] | 95% CI | | P |
| Hydralazine <180 days | Reference | | | | Reference | | | |
| Hydralazine ≥180 days | 0.762 | 0.653 | 0.897 | <.001 | 0.789 | 0.667 | 0.913 | <.001 |
| **Sex** | | | | | | | | |
| Male | 1.185 | 0.893 | 1.886 | .258 | 1.246 | 0.910 | 1.962 | .240 |
| Female | Reference | | | | Reference | | | |
| **Age group (yr)** | | | | | | | | |
| 20–29 | Reference | | | | Reference | | | |
| 30–39 | 1.158 | 0.659 | 1.395 | .778 | 1.299 | 0.389 | 1.894 | .738 |
| 40–49 | 1.122 | 0.528 | 1.327 | .852 | 1.194 | 0.233 | 1.731 | .822 |
| 50–59 | 1.119 | 0.541 | 1.351 | .839 | 1.205 | 0.239 | 1.753 | .814 |
| ≥60 | 1.173 | 0.675 | 1.404 | .584 | 1.321 | 0.397 | 1.923 | .747 |
| **Insured premium (NTD)** | | | | | | | | |
| <18,000 | Reference | | | | Reference | | | |
| 18,000–34,999 | 1.069 | 0.726 | 1.731 | .411 | 1.102 | 0.750 | 1.756 | .392 |
| ≥35,000 | 0.792 | 0.484 | 1.186 | .604 | 0.894 | 0.500 | 1.250 | .579 |
| Normal pregnancy | 0.894 | 0.500 | 1.145 | .487 | 0.826 | 0.478 | 1.057 | .499 |
| **Comorbidities** (Reference: Without) | | | | | | | | |
| CHF | 0.955 | 0.710 | 1.185 | 0.397 | 1.274 | 1.059 | 1.571 | .030 |
| PE | 1.143 | 0.857 | 1.438 | .189 | 1.404 | 1.006 | 1.655 | .079 |
| GI hemorrhage | 1.209 | 0.708 | 1.617 | .384 | 1.497 | 1.006 | 2.195 | .081 |
| Cerebral thrombosis | 1.035 | 0.553 | 1.142 | .501 | 1.133 | .749 | 1.250 | .295 |
| IHD | 1.642 | 1.175 | 2.047 | <.001 | 1.952 | 1.509 | 2.377 | <.001 |
| Vascular insufficiency of intestine | 1.165 | 1.032 | 1.648 | .035 | 1.607 | 1.174 | 2.054 | <.001 |
| Obesity | 1.430 | 0.214 | 2.040 | .755 | 1.662 | 0.375 | 2.799 | .686 |
| HBV with hepatic coma | 2.652 | 1.762 | 3.487 | <.001 | 3.024 | 1.802 | 3.592 | <.001 |
| HBV without hepatic coma | 2.101 | 1.356 | 2.977 | <.001 | 2.256 | 1.450 | 2.986 | <.001 |
| Malignant neoplasm of kidney/renal pelvis | 1.619 | 1.073 | 1.950 | .004 | 1.702 | 1.085 | 1.996 | .001 |
| Acute glomerulonephritis/ Nephrotic syndrome | 1.642 | 1.133 | 2.143 | <.001 | 2.211 | 1.507 | 3.929 | <.001 |
| Proteinuria | 1.191 | 0.831 | 1.583 | .298 | 1.307 | 0.915 | 1.653 | .210 |
| Gestational hypertension | 1.478 | 0.962 | 2.309 | .090 | 1.742 | 0.992 | 2.657 | .058 |
| Asthma | 1.515 | 0.811 | 2.101 | .307 | 1.614 | 0.863 | 2.181 | .295 |
| CCI_R | 1.483 | 1.350 | 1.630 | <.001 | 1.692 | 1.571 | 1.834 | <.001 |
| **Medications** (Reference: Without) | | | | | | | | |
| Aspirin | 1.575 | 1.078 | 2.053 | .039 | 1.653 | 1.101 | 2.132 | .014 |
| Celecoxib | 1.513 | 1.071 | 1.846 | .034 | 1.608 | 1.163 | 1.933 | <.001 |
| Itraconazole | 0.792 | 0.314 | 1.653 | .726 | 0.927 | 0.385 | 1.690 | .698 |
| Mebendazole | 0.582 | 0.180 | 0.936 | <.001 | 0.696 | 0.286 | 0.957 | .003 |
| Leflunomide | 1.290 | 0.831 | 1.564 | .337 | 1.385 | 0.871 | 1.651 | .309 |
| Thalidomide | 1.565 | 1.136 | 2.184 | <.001 | 1.846 | 1.194 | 2.466 | <.001 |
| Valproate | 1.089 | 0.364 | 1.347 | .815 | 1.475 | 0.703 | 2.180 | .674 |
| Metformin | 0.869 | 0.671 | 1.037 | .264 | 0.930 | 0.728 | 1.143 | .189 |
| Auranofin | 1.163 | 0.699 | 1.738 | .0385 | 1.366 | 0.847 | 1.904 | .293 |
| Statins | 1.268 | 0.817 | 1.867 | .238 | 1.633 | 1.004 | 2.158 | .044 |
| Bisphosphonates | 1.105 | 0.595 | 1.655 | .540 | 1.230 | 0.264 | 1.731 | .502 |
| Bromocriptine | 1.263 | 0.654 | 1.896 | 0.384 | 1.301 | 0.659 | 1.996 | .379 |

*(Continued)*

**Table 2.** (Continued)

| | No competing risk model | | | | Fine and Gray's competing risk model[†] | | | |
|---|---|---|---|---|---|---|---|---|
| Chlorprothixene | 1.146 | 0.452 | 1.975 | 0.662 | 1.245 | 0.482 | 2.030 | .656 |
| Clotrimazole | 1.896 | 0.597 | 2.340 | 0.480 | 1.962 | 0.633 | 2.385 | .471 |
| Quinacrine | 1.036 | 0.716 | 1.852 | 0.367 | 1.076 | 0.725 | 1.986 | .325 |
| Ivermectin | 1.745 | 0.389 | 2.870 | 0.462 | 1.753 | 0.401 | 2.901 | .448 |
| Verteporfin | 1.482 | 0.893 | 1.997 | 0.152 | 1.502 | 0.899 | 2.131 | .130 |
| Clarithromycin | 1.207 | 0.131 | 1.585 | 0.903 | 1.284 | 0.176 | 1.627 | .897 |
| Hydroxychloroquine | 0.986 | 0.255 | 1.264 | 0.686 | 1.030 | 0.579 | 1.345 | .650 |
| Tofacitinib | 1.335 | 0.797 | 1.801 | 0.335 | 1.348 | 0.803 | 1.829 | .302 |
| Gefitinib | 1.124 | 0.543 | 1.675 | 0.452 | 1.166 | 0.552 | 1.388 | .428 |
| Curcumin | 1.088 | 0.670 | 1.337 | 0.381 | 1.127 | 0.668 | 1.350 | .375 |
| Chlorhexidine | 1.297 | 0.884 | 1.509 | 0.234 | 1.319 | 0.897 | 1.573 | .208 |
| Axitinib | 1.303 | 1.000 | 1.525 | 0.050 | 1.325 | 1.026 | 1.599 | .024 |
| **Season of index date** | | | | | | | | |
| Spring | Reference | | | | Reference | | | |
| Summer | 0.745 | 0.493 | 1.185 | .528 | 0.912 | 0.723 | 1.259 | .511 |
| Autumn | 0.575 | 0.405 | 1.124 | .696 | 0.791 | 0.650 | 1.223 | .684 |
| Winter | 0.826 | 0.664 | 1.420 | .347 | 0.890 | 0.678 | 1.529 | .325 |
| **Urbanization level** | | | | | | | | |
| 1 (The highest) | 1.356 | 0.826 | 1.859 | .294 | 1.396 | 0.839 | 1.865 | .204 |
| 2 | 1.229 | 0.708 | 1.785 | .385 | 1.290 | 0.749 | 1.826 | .298 |
| 3 | 1.130 | 0.622 | 1.748 | .465 | 1.196 | 0.679 | 1.771 | .326 |
| 4 (The lowest) | Reference | | | | Reference | | | |
| **Levels of hospitals** | | | | | | | | |
| Medical center | 1.704 | 1.331 | 2.171 | <.001 | 2.470 | 2.065 | 2.905 | <.001 |
| Regional hospital | 1.505 | 1.175 | 1.852 | <.001 | 2.083 | 1.645 | 2.475 | <.001 |
| Local hospital | Reference | | | | Reference | | | |

All variables controlled by the models (‡ and §) include demographics (sex, age, insured premium, location, urbanization level, and level of hospital), comorbidities (congestive heart failure, pulmonary embolism, gastrointestinal hemorrhage, cerebral thrombosis, ischemic heart disease, vascular insufficiency of intestine, obesity, malignant neoplasm of kidney/renal pelvis, acute glomerulonephritis/nephrotic syndrome, proteinuria, gestational hypertension, and asthma), other variables (normal pregnancy and Charlson Comorbidity Index_Revised), and medications (aspirin, celecoxib, itraconazole, mebendazole, leflunomide, thalidomide, valproate, metformin, auranofin, statins [nystatin, lovastatin, pravastatin, simvastatin, atorvastatin, pitavastatin, rosuvastatin, cilastatin], bisphosphonates [alendronate and risedronate], bromocriptine, chlorprothixene, clotrimazole, quinacrine, ivermectin, verteporfin, clarithromycin, hydroxychloroquine, tofacitinib, gefitinib, curcumin, chlorhexidine, and axitinib).

*Proportional-hazards assumption test was checked based on Schoenfeld residuals. Global test: $P=0.8947$ (without competing), $P=0.8835$ (with competing).

[†]Competing variable was all-cause mortality.

[‡]Adjusted HR, adjusted hazard ratio.

[§]Adjusted sHR, adjusted subdistribution hazard ratio.

NTD, New Taiwan dollar; CHF, congestive heart failure; PE, pulmonary embolism; GI, gastrointestinal; IHD, ischemic heart disease; HBV, hepatitis B virus; CCI_R, Charlson Comorbidity Index_Revised; HR, hazard ratio; CI, confidence interval.

A statistically significant, inverse duration-response association was observed between hydralazine use and hematologic neoplasm risk across several subgroups. This association was most pronounced for other malignant neoplasms of lymphoid and histiocytic tissue, MM and immunoproliferative neoplasms, and other polycythemia; these key findings were presented in Table 3. The detailed analyses for all other hematologic neoplasm subgroups were provided in Table E in S1 File.

**Table 3. Adjusted hazard ratio of hematologic neoplasm development for overall risk and key subgroups, stratified by prescription duration of hydralazine.**

| Subgroups of hematologic neoplasms | Prescription duration of hydralazine | Population | Events | PYs | Rate (per $10^5$ PYs) | Adjusted HR‡ | 95% CI | | P |
|---|---|---|---|---|---|---|---|---|---|
| Overall | <180 days | 239,144 | 4,544 | 2,834,197.06 | 160.33 | Reference | | | |
| | ≥180 days | 59,786 | 757 | 716,983.56 | 105.58 | 0.762 | 0.653 | 0.897 | <.001 |
| | 180–350 days | 19,868 | 294 | 238,267.67 | 123.39 | 0.884 | 0.743 | 1.098 | .189 |
| | 351–667 days | 19,975 | 245 | 239,805.11 | 102.17 | 0.728 | 0.598 | 0.904 | <.001 |
| | ≥668 days | 19,943 | 218 | 238,910.78 | 91.25 | 0.646 | 0.531 | 0.803 | <.001 |
| Other malignant neoplasms of lymphoid and histiocytic tissue | <180 days | 239,144 | 541 | 2,834,197.06 | 19.09 | Reference | | | |
| | ≥180 days | 59,786 | 68 | 716,983.56 | 9.48 | 0.558 | 0.459 | 0.694 | <.001 |
| | 180–350 days | 19,868 | 28 | 238,267.67 | 11.75 | 0.722 | 0.593 | 0.896 | <.001 |
| | 351–667 days | 19,975 | 21 | 239,805.11 | 8.76 | 0.516 | 0.424 | 0.641 | <.001 |
| | ≥668 days | 19,943 | 19 | 238,910.78 | 7.95 | 0.440 | 0.361 | 0.545 | <.001 |
| Multiple myeloma and immunoproliferative neoplasms | <180 days | 239,144 | 369 | 2,834,197.06 | 13.02 | Reference | | | |
| | ≥180 days | 59,786 | 53 | 716,983.56 | 7.39 | 0.598 | 0.492 | 0.743 | <.001 |
| | 180–350 days | 19,868 | 19 | 238,267.67 | 7.97 | 0.656 | 0.538 | 0.814 | <.001 |
| | 351–667 days | 19,975 | 18 | 239,805.11 | 7.51 | 0.595 | 0.489 | 0.739 | <.001 |
| | ≥668 days | 19,943 | 16 | 238,910.78 | 6.70 | 0.544 | 0.447 | 0.676 | <.001 |
| Other polycythemia | <180 days | 239,144 | 1,722 | 2,834,197.06 | 60.76 | Reference | | | |
| | ≥180 days | 59,786 | 303 | 716,983.56 | 42.26 | 0.790 | 0.649 | 0.897 | <.001 |
| | 180–350 days | 19,868 | 103 | 238,267.67 | 43.23 | 0.806 | 0.662 | 0.924 | <.001 |
| | 351–667 days | 19,975 | 101 | 239,805.11 | 42.12 | 0.785 | 0.646 | 0.854 | <.001 |
| | ≥668 days | 19,943 | 99 | 238,910.78 | 41.44 | 0.778 | 0.640 | 0.823 | <.001 |

All variables controlled by the model (‡) include demographics (sex, age, insured premium, location, urbanization level, and level of hospital), comorbidities (congestive heart failure, pulmonary embolism, gastrointestinal hemorrhage, cerebral thrombosis, ischemic heart disease, vascular insufficiency of intestine, obesity, malignant neoplasm of kidney/renal pelvis, acute glomerulonephritis/nephrotic syndrome, proteinuria, gestational hypertension, and asthma), other variables (normal pregnancy and Charlson Comorbidity Index_Revised), and medications (aspirin, celecoxib, itraconazole, mebendazole, leflunomide, thalidomide, valproate, metformin, auranofin, statins [nystatin, lovastatin, pravastatin, simvastatin, atorvastatin, pitavastatin, rosuvastatin, cilastatin], bisphosphonates [alendronate and risedronate], bromocriptine, chlorprothixene, clotrimazole, quinacrine, ivermectin, verteporfin, clarithromycin, hydroxychloroquine, tofacitinib, gefitinib, curcumin, chlorhexidine, and axitinib).

‡Adjusted HR, adjusted hazard ratio.

PYs, person-years; HR, hazard ratio; CI, confidence interval.

This association suggested a duration-dependent pattern, with a greater reduction in risk observed with longer prescription durations, a finding that was consistent across both models (Table C in S1 File). The median duration of hydralazine prescription was 9.18 years (Table G in S1 File), and the corresponding data exhibited an approximately normal distribution. Notably, hydralazine was associated with a lower risk of subsequent development of leukemia of unspecified cell type, as shown in the multiple-event model (Table C in S1 File).

**Leave-one-out analysis stratified by hematologic neoplasm subgroup and duration of hydralazine prescription**

To confirm that the observed inverse association was not disproportionately driven by the "other polycythemia" subgroup, which constituted 38% of all hematologic neoplasm cases, we performed a leave-one-out sensitivity analysis. Accordingly, an additional subgroup analysis was performed to determine the composition of other polycythemia. As displayed in Table

I in S1 File, 2,025 individuals received a diagnosis of other polycythemia over the 16-year follow-up period, of whom 2,005 had secondary polycythemia (90.9%) and 20 had familial polycythemia (0.99%). The leave-one-out analysis (Table F in S1 File) demonstrated the statistical robustness of the finding, as the inverse association between hydralazine use and hematologic neoplasms persisted after excluding cases of secondary polycythemia.

### Association of long-term hydralazine use with hematologic neoplasm incidence

Long-term hydralazine use was associated with a longer time to hematologic neoplasm diagnosis. The median interval from the index date to the first hematologic neoplasm diagnosis was significantly longer in the exposure group than in the reference group (7.33 versus 7.05 years; $P<.001$; Table H in S1 File). After the 16-year follow-up, hematologic neoplasm had been diagnosed in 757 patients (1.27%) in the exposure group and 4,544 (1.90%) in the reference group (Table I in S1 File). Compared with that in the reference group, the adjusted HRs of overall hematologic neoplasm incidence in the exposure group after adjustment for covariates and comorbidities were 0.762 (95% confidence interval [0.653,0.897]; $P<.001$; Table 3). Additionally, the exposure group exhibited a significantly lower cumulative incidence of overall hematologic neoplasm (log-rank $P$ value in the 16th year being $<.001$; Fig 2A) than did the reference group. This pattern was also seen in the analysis of cumulative incidence, which was significantly lower in the exposure group for several hematologic neoplasm subgroups, including MM and immunoproliferative neoplasm (Fig 2B), myeloid leukemia (Fig 2C), neoplasm of uncertain behavior (Fig 2D), and other polycythemia (Fig 2E) (all log-rank $P<.001$).

### Sensitivity analysis for the hematologic neoplasm incidence

To minimize potential selection bias arising from the inclusion of patients with ongoing hematologic neoplasm in the early stages of follow-up, a sensitivity analysis was performed by excluding patients diagnosed within the first year or the first 5 years. Table 4 presents the results of this analysis. After excluding patients diagnosed within the first year, the association for the overall ≥180 days group was no longer statistically significant (adjusted sHR = 0.787, 95% confidence interval [0.646,0.980]; $P=.03$). Similarly, after excluding the first five years, the association for this group did not meet our pre-specified threshold (adjusted sHR = 0.798, 95% CI [0.659,0.950]; $P=.001$). However, the duration-dependent pattern remained robust; the association for the longest-duration subgroup (≥668 days) remained statistically significant in both the 1-year exclusion (adjusted sHR = 0.674, 95% CI [0.555,0.838]; $P<.001$) and 5-year exclusion (adjusted sHR = 0.684, 95% CI [0.565,0.858]; $P<.001$) analyses.

### Mortality analysis

The mortality analysis showed that there was no significant difference in hematologic neoplasm-related mortality between the exposure group and reference group (adjusted HR = 0.884, 95% confidence interval [0.632,1.238]; $P=.265$, Table J in S1 File). Similarly, all-cause mortality was not significantly different between the two groups (adjusted HR = 1.075, 95% CI [0.768,1.506]; $P=.536$). These results indicate that hydralazine use was associated with a lower risk of developing hematologic neoplasm but did not significantly alter long-term survival outcomes. The absence of an observed difference in mortality, despite the lower risk of hematologic neoplasms in the exposure group, might be attributable to the higher baseline comorbidity burden in these patients. This higher burden could have masked any potential association between a lower risk of hematologic neoplasms and survival.

## Discussion

This nationwide retrospective cohort study investigated the association between the use of AHAs and the risk of developing hematologic neoplasms in patients with hypertension. Our findings indicate that the exposure group had an approximately 21% lower risk of overall hematologic neoplasm than did the reference group, particularly for other malignant neoplasms of lymphoid and histiocytic tissue (43% lower), MM and immunoproliferative neoplasms (38% lower), and

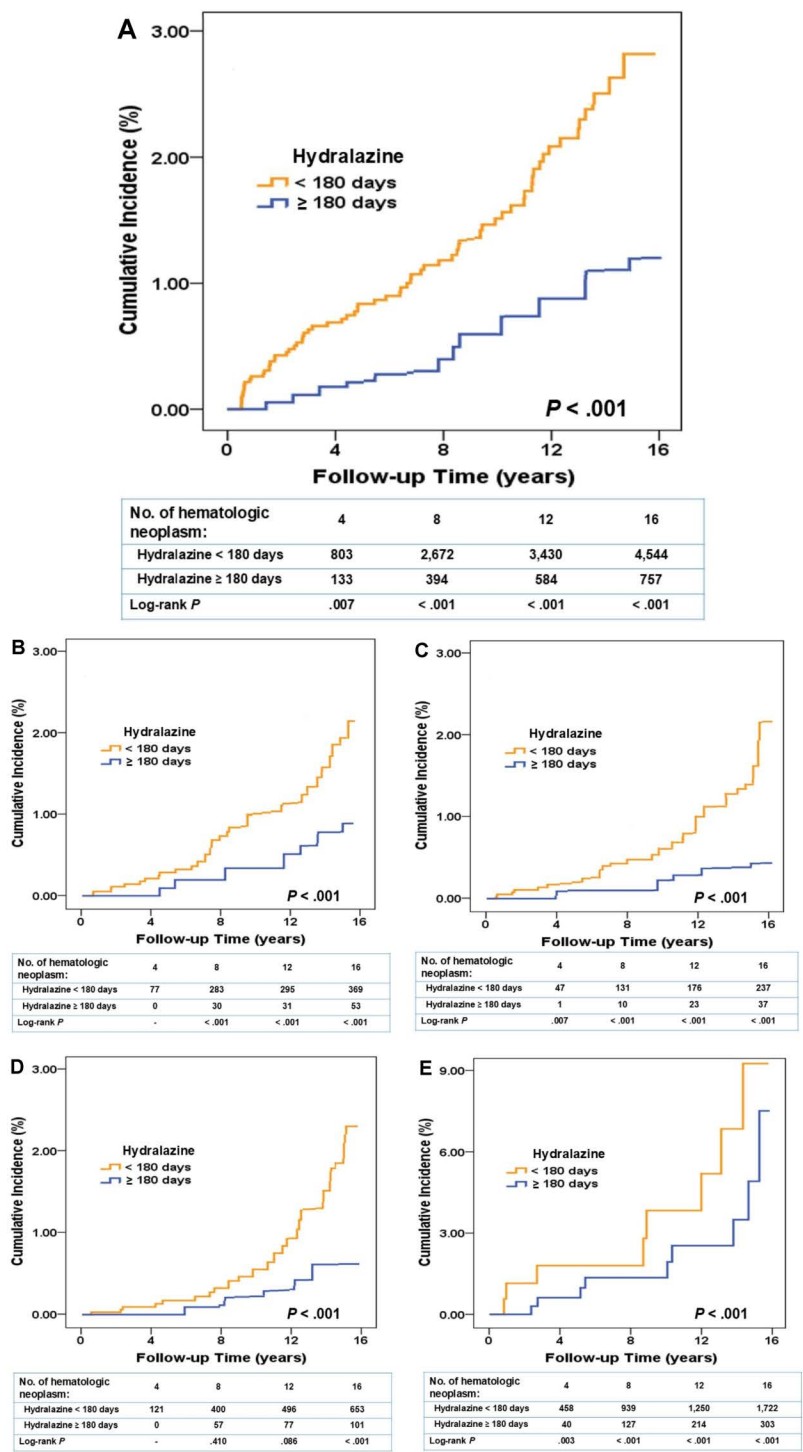

**Fig 2. Association of long-term hydralazine use with the incidence of hematologic neoplasms in patients with hypertension, 2000–2015.** The cumulative incidence of hematologic neoplasms in the reference group (hydralazine exposure <180 days) is indicated by the orange line, and that in the exposure group (hydralazine exposure ≥180 days) is indicated by the blue line. **(A)** Cumulative incidence of overall hematologic neoplasm was significantly lower. **(B–E)** Cumulative incidence stratified by hematologic neoplasm subgroups was significantly lower in the following subgroups: (B) multiple myeloma and immunoproliferative neoplasm, (C) myeloid leukemia(), (D) neoplasms of uncertain behavior, and (E) other polycythemia.

**Table 4. Sensitivity analysis for factors of hematologic neoplasm development by using Cox regression in competing risk model.**

| Sensitivity analysis | Prescription duration of hydralazine | Popula-tions | Events | PYs | Rate (per 10⁵ PYs) | No competing risk model | | | Fine and Gray's competing risk model* | | |
|---|---|---|---|---|---|---|---|---|---|---|---|
| | | | | | | Adjusted HR‡ | 95% CI | P | Adjusted sHR§ | 95% CI | P |
| Overall | <180 days | 239,144 | 4,544 | 2,834,197.06 | 160.33 | Reference | | | Reference | | |
| | ≥180 days | 59,786 | 757 | 716,983.56 | 105.58 | 0.762 | 0.653 0.897 | <.001 | 0.789 | 0.667 0.913 | <.001 |
| | 180–350 days | 19,868 | 294 | 238,267.67 | 123.39 | 0.884 | 0.743 1.098 | .189 | 0.916 | 0.767 1.134 | .142 |
| | 351–667 days | 19,975 | 245 | 239,805.11 | 102.17 | 0.728 | 0.598 0.904 | <.001 | 0.754 | 0.618 0.935 | <.001 |
| | ≥668 days | 19,943 | 218 | 238,910.78 | 91.25 | 0.646 | 0.531 0.803 | <.001 | 0.666 | 0.552 0.831 | <.001 |
| In the first year excluded | <180 days | 239,144 | 4,260 | 2,657,043.24 | 160.33 | Reference | | | Reference | | |
| | ≥180 days | 59,786 | 711 | 672,174.25 | 105.78 | 0.759 | 0.624 0.972 | .028 | 0.787 | 0.646 0.98 | .03 |
| | 180–350 days | 19,868 | 276 | 223,378.13 | 123.56 | 0.887 | 0.729 1.103 | .333 | 0.918 | 0.756 1.148 | .304 |
| | 351–667 days | 19,975 | 235 | 224,897.20 | 104.49 | 0.736 | 0.605 0.915 | <.001 | 0.762 | 0.623 0.947 | <.001 |
| | ≥668 days | 19,943 | 200 | 223,898.92 | 89.33 | 0.652 | 0.538 0.812 | <.001 | 0.674 | 0.555 0.838 | <.001 |
| In the first 5 years excluded | <180 days | 239,144 | 3,115 | 1,948,522.27 | 159.86 | Reference | | | Reference | | |
| | ≥180 days | 59,786 | 517 | 492,464.35 | 104.98 | 0.771 | 0.636 0.957 | .004 | 0.798 | 0.659 0.95 | .001 |
| | 180–350 days | 19,868 | 203 | 163,798.24 | 123.93 | 0.901 | 0.723 1.093 | .246 | 0.935 | 0.743 1.132 | .237 |
| | 351–667 days | 19,975 | 166 | 164,863.25 | 100.69 | 0.745 | 0.613 0.927 | <.001 | 0.77 | 0.632 0.946 | <.001 |
| | ≥668 days | 19,943 | 148 | 163,802.86 | 90.35 | 0.666 | 0.548 0.828 | <.001 | 0.684 | 0.565 0.858 | <.001 |

All variables controlled by the models (‡ and §) include demographics (sex, age, insured premium, location, urbanization level, and level of hospital), comorbidities (congestive heart failure, pulmonary embolism, gastrointestinal hemorrhage, cerebral thrombosis, ischemic heart disease, vascular insufficiency of intestine, obesity, malignant neoplasm of kidney/renal pelvis, acute glomerulonephritis/nephrotic syndrome, proteinuria, gestational hypertension, and asthma), other variables (normal pregnancy and Charlson Comorbidity Index_Revised), and medications (aspirin, celecoxib, itraconazole, mebendazole, leflunomide, thalidomide, valproate, metformin, auranofin, statins [nystatin, lovastatin, pravastatin, simvastatin, atorvastatin, pitavastatin, rosuvastatin, cilastatin], bisphosphonates [alendronate and risedronate], bromocriptine, chlorprothixene, clotrimazole, quinacrine, ivermectin, verteporfin, clarithromycin, hydroxychloroquine, tofacitinib, gefitinib, curcumin, chlorhexidine, and axitinib).*Competing variable was all-cause mortality.

‡Adjusted HR, adjusted hazard ratio.

§Adjusted sHR, adjusted subdistribution hazard ratio.

PYs, person-years; HR, hazard ratio; CI, confidence interval.

myeloid leukemia (29% lower). Furthermore, multivariable analysis revealed a duration-dependent inverse association between hydralazine use and the risk of hematologic neoplasms in patients with hypertension.

Cardiovascular disease (CVD) and cancers are the top 2 leading causes of mortality worldwide [48]. Notably, hypertension is not only the leading cause of CVD [49] but also a risk factor for various types of cancer; hence, determining an optimal strategy for hypertension management could reduce global mortality. A nationwide cohort study [2] and the current study both indicated a relatively high risk of hematologic neoplasm development in patients with hypertension, signifying that hypertension is associated not only with CVD and solid tumors but also with hematologic neoplasms. Although the association between the use of AHAs and the risk of certain solid tumors has been explored—with their involvement in biological functions such as reducing inflammation and angiogenesis being documented in preclinical studies [9]—the specific mechanisms underlying a potential association with hematologic neoplasms remain largely unclear, and further clinical investigation is needed.

In the present study, long-term hydralazine use was associated with a significantly lower risk of hematologic neoplasms across several subgroups; this finding suggests the association may be linked to several biological regulatory mechanisms. In addition to its antihypertensive action, hydralazine targets DNMT [50], enhances P53 function [51], and

participates in other crucial anti-hematologic neoplasm signaling pathways. Hydralazine has been demonstrated to reduce the viability of monocytic leukemia cells [52] and to counteract chemoresistance in chronic myeloid leukemia [53], a clonal disorder associated with the Philadelphia chromosome, which results from the t(9;22) translocation and carries the BCR-ABL fusion gene that encodes the oncogenic BCR-ABL protein. This chimeric protein leads to the aberrant activation of several signaling pathways, including the PI3K-AKT pathway. Hydralazine has also been suggested to inhibit AKT activation (as observed in an animal sepsis model [54]) and to reduce cleaved caspase-3 and caspase-9 levels (as shown in a rat model of cardiac injury [55]), and has been demonstrated to promote caspase-dependent apoptotic cell death in human leukemic T cells [17]. Moreover, hydralazine inhibits glutamic-oxaloacetic transaminase 1 (a finding from an in vitro screening assay [56]), a prognostic marker of AML [57], indicating its potential role in reducing AML risk. Hydralazine also inhibits angiogenesis (a finding from both in vitro and animal studies [58]) by suppressing vascular endothelial growth factor and basic fibroblast growth factor signaling, both of which are correlated with the clinicopathological features of myeloproliferative neoplasms [59] and MM [60]. Overall, in addition to its role in managing hypertension, hydralazine use was associated with a lower risk of hematologic neoplasms. This association may be explained by its activity in several biological pathways.

Conversely, studies exploring the association between the use of NSAIDs—such as aspirin [61], celecoxib [62], and thalidomide [63]—and the risk of hematologic neoplasms have reported inconsistent findings. For example, some studies have described an association with a decreased risk of certain hematologic neoplasms, others have found no association [64], while some have observed an association with an increased risk [65]. Most NSAIDs have been reported to interfere with the therapeutic action of AHAs [66], an interaction that may be associated with increased blood pressure. This proposed mechanism may help explain the association with a higher risk of hematologic neoplasms that was observed for celecoxib and thalidomide in our 16-year follow-up cohort study. While the association between aspirin, a unique NSAID, and blood pressure remains controversial, some cohort studies have reported that aspirin use is linked to an increased risk of developing hypertension [67]. It remains unclear whether aspirin increases or decreases cancer risk. A meta-analysis of cohort studies [68] revealed that when used at low doses, aspirin can reduce the risk of colorectal cancer, but at high doses, it can increase the risks of lung cancer and prostate cancer. Additionally, aspirin was reported to accelerate the progression of both solid cancers and HMs in older adults [69]. The discrepancies in the reported association between aspirin use and cancer risk may be attributable to methodological heterogeneity across studies, such as variations in dosage, study populations, or the specific cancer types investigated. In the present cohort study, more than 50% of the enrolled patients were aged ≥60 years (Table 1) and exhibited a higher incidence of hypertension and lower immune surveillance, which resulted in an increased hematologic neoplasm risk. Therefore, considering the elevated baseline risk in this older population with hypertension, our findings—which include an observed association between certain NSAIDs and an increased risk of hematologic neoplasms—do not support a potential risk-reducing role for these agents in this context.

Axitinib, a tyrosine kinase inhibitor with antileukemic activity [70], has been associated with the induction and exacerbation of hypertension [71], which may contribute to a higher risk of hematologic neoplasms; this finding was consistent with our study findings (adjusted HR = 1.303, 95% confidence interval [1.000,1.525]; P = .005). Notably, the broad-spectrum anthelmintic medication mebendazole was associated with a lower risk of hematologic neoplasm development in this study. Mebendazole has previously been reported to inhibit the growth of various AML cell lines and mononuclear cells derived from the bone marrow of patients with AML in vitro. This inhibitory effect is thought to be mediated by the downregulation of Akt and Erk signaling pathways [43]. However, to our knowledge, no cohort study has reported an association between mebendazole use and the risk of leukemia in patients with hypertension.

Although the association between hypertension and a higher risk of several cancers has been established, evidence regarding the effect of AHAs on cancer risk is inconsistent [72]. The potential anticancer efficacy of AHAs may be diminished by the interferences resulting from simultaneous multidrug interactions. Furthermore, cohort studies with insufficient

control for confounders are prone to spurious associations that can mask, or even invert, the true relationship between an AHA and cancer risk.

A critical consideration for these findings, however, is the known safety profile of long-term hydralazine use. The potential for dose-dependent adverse effects, such as hydralazine-associated lupus-like adverse effects (HAAEs), raises major concerns. According to a previous cohort study involving 36,349 patients with hypertension [2], a daily dose of <34 mg was associated with a significantly lower risk of overall hematologic neoplasm (adjusted HR = 0.791, 95% confidence interval [0.578,0.927]; $P < .001$) when compared with hydralazine non-users. However, no case of HAAEs was reported in patients receiving a daily dose of 50 mg [73], suggesting that the dosage (<34 mg per day) associated with a lower risk of hematologic neoplasm was considerably below the dosage at which HAAEs have been observed. Although slow acetylators are generally considered more susceptible to HAAEs [74], HAAEs have rarely been reported among patients receiving a daily hydralazine dose of <50 mg, regardless of their acetylator status.

Our study has several strengths, including its large sample size and its use of verified information for evaluating long-term hydralazine-associated hematologic neoplasm risks. However, this study has some limitations that should be considered. First, potential misinformation may have arisen from errors in the NHIRD. Second, the LGTD lacks data on key behavioral and socioeconomic confounders. Information on lifestyle factors such as smoking, alcohol consumption, and physical activity, as well as formal socioeconomic status indicators beyond insurance premiums, was unavailable for adjustment. Third, the relationship between hematologic neoplasm severity and hypertension was not evaluated. Fourth, the lack of available laboratory data limited our ability to identify the potential mechanisms underlying the observed association between hydralazine use and hematologic neoplasm development. Fifth, the study did not include several AHAs (spironolactone, α-blockers, and β-blockers) for comparison with hydralazine, potentially introducing bias. However, the antineoplastic efficacy of these agents for hematologic neoplasms is not yet reported, except for prazosin [75]. Finally, we could not directly contact patients to verify their use of hydralazine and medication compliance due to their anonymous identities. Although some patients with hypertension may have had poor medication adherence, our consideration of a prescription period of ≥180 days may have minimized this potential bias. Accordingly, the observed association between hydralazine use and a lower risk of hematologic neoplasm development remained, despite the possible underestimation of the actual dosage of hydralazine. This finding suggests that the association between hydralazine and a lower risk of hematologic neoplasm may be relevant in real-world clinical practice.

Despite these limitations, we believe that our retrospective study provides real-world evidence and valuable insights into the association between the use of antihypertensive hydralazine and the risk of hematologic neoplasms in patients with hypertension. Although the observational design cannot establish causality, the proposed association is supported by several credible findings, including a duration-dependent relationship and consistent results across multiple analyses. Clinically, the combination of hydralazine and valproate has shown activity in the treatment of MDS [19] and cutaneous T-cell lymphoma [76]. This clinical observation provides a parallel to the association found in our study between hydralazine use and a lower risk of hematologic neoplasms. Furthermore, to check for selection bias, we compared baseline demographic and clinical characteristics between exposure group and those excluded during initial screening. The excluded individuals were clinically distinct but had been removed prior to propensity score matching. Specifically, our leave-one-out analysis indicated that the association between hydralazine use and a lower risk of hematologic neoplasm persisted even after excluding cases from a major hematologic neoplasm subgroup, supporting the robustness of the observed association for overall hematologic neoplasms. These methodological approaches ensured that the included cohorts were well-balanced, thereby minimizing selection bias. In conclusion, our results highlight hydralazine as a compelling candidate for drug repurposing to address the risk of hematologic neoplasms. Such a strategy is advantageous because it may circumvent the protracted timelines and substantial costs inherent in novel drug development.

It is crucial to consider our findings within the clinical context of hydralazine use. As a later-line AHA, hydralazine is often prescribed to patients with more severe or refractory hypertension, heart failure, or chronic kidney disease. Our

baseline data reflected this reality, as the exposure group had a significantly higher burden of cardiovascular and renal comorbidities, including CHF, IHD, and glomerulonephritis, as well as higher CCI_R scores (Table 1). This confounding by indication would typically bias the results towards an increased risk of adverse outcomes in the hydralazine group. Therefore, the observation of a significantly lower risk of hematologic neoplasms in the group with a higher comorbidity burden—an association that persisted after multivariable adjustment—strengthens the robustness of our findings. For patients with an existing indication, hydralazine use may be associated with a lower risk of hematologic neoplasm. Nonetheless, prospective studies are warranted to further investigate this association and its potential clinical implications.

In conclusion, our findings indicate that the use of hydralazine in patients with hypertension is associated with a significantly reduced risk of hematologic neoplasms. This association suggests that for patients with hypertension, particularly those with multiple susceptibility factors for hematologic neoplasms, hydralazine use may be linked to a lower incidence of these neoplasms, warranting further prospective studies to investigate this relationship.

## Supporting information

**S1 Checklist. This checklist is provided in accordance with the STROBE statement, available from https://www.strobe-statement.org/.**
(DOCX)

**S1 File. Table A**. Baseline characteristics of the hydralazine cohort compared to patients with hypertension excluded during initial screening. **Table B.** ICD-9-CM coding and definition. **Table C.** Comparison of the adjusted subdistribution hazard ratio of hematologic neoplasms according to subgroup stratified by prescription duration of hydralazine in first-event and multiple-event models in a competing risk model. **Table D.** Multivariable risk regression analysis of hematologic neoplasm development in patients without/with hypertension in competing risk model. **Table E.** Adjusted hazard ratio for remaining hematologic neoplasm subgroups, stratified by prescription duration of hydralazine. **Table F.** Leave-one-out analysis for comparison of adjusted hazard ratio of hematologic neoplasms according to subgroup stratified by prescription duration of hydralazine in first-event and multiple-event models in a competing risk model. **Table G.** Tracking years in patients with hypertension by prescription duration of hydralazine. **Table H.** Tracking years from initiating hydralazine prescription to having hematologic neoplasms in patients with hypertension. **Table I.** Endpoint characteristics of patients with hypertension by prescription duration of hydralazine, 2000–2015. **Table J.** Mortality analysis of patients with hypertension by prescription duration of hydralazine, 2000–2015. **Table K.** Unadjusted (crude) hazard ratios for risk factors associated with hematologic neoplasm development. **Table L.** Unadjusted (crude) hazard ratios for hematologic neoplasm development, stratified by prescription duration of hydralazine. **Table M.** Unadjusted (crude) hazard ratios for sensitivity analysis of hematologic neoplasm development. **Table N.** Unadjusted (crude) subdistribution hazard ratios for first-event and multiple-event models. **Table O.** Unadjusted (crude) hazard ratios for hematologic neoplasm risk associated with hypertension. **Table P.** Unadjusted (crude) subdistribution hazard ratios for leave-one-out sensitivity analysis. **Table Q.** Unadjusted (crude) hazard ratios for mortality analysis.
(DOCX)

## Acknowledgments

The authors thank Wallace Academic Editing (www.editing.tw/) and Editage (www.editage.com.tw) for editing the manuscript.

## Disclaimer

This study is based in part on data from the National Health Insurance Research Database provided by the Bureau of National Health Insurance, Department of Health, and managed by National Health Research Institutes.

## Author contributions

**Conceptualization:** Wu-Chien Chien, Chi-Hsiang Chung, Yeu-Chin Chen, Wei-Che Tsai, Bing-Heng Yang.

**Data curation:** Li-Tzu Wang, Wu-Chien Chien, Kevin Sheng-Kai Ma, Chi-Hsiang Chung, Bing-Heng Yang.

**Formal analysis:** Li-Tzu Wang, Kevin Sheng-Kai Ma, Chi-Hsiang Chung, Bing-Heng Yang.

**Funding acquisition:** Li-Tzu Wang, Wu-Chien Chien, Bing-Heng Yang.

**Investigation:** Li-Tzu Wang, Kevin Sheng-Kai Ma, Chi-Hsiang Chung.

**Methodology:** Wu-Chien Chien, Chi-Hsiang Chung, Yeu-Chin Chen, Wei-Che Tsai, Bing-Heng Yang.

**Project administration:** Wu-Chien Chien, Chi-Hsiang Chung, Bing-Heng Yang.

**Resources:** Wu-Chien Chien, Chi-Hsiang Chung, Bing-Heng Yang.

**Supervision:** Wu-Chien Chien, Bing-Heng Yang.

**Validation:** Bing-Heng Yang.

**Visualization:** Li-Tzu Wang, Wu-Chien Chien, Kevin Sheng-Kai Ma, Chi-Hsiang Chung, Yeu-Chin Chen, Wei-Che Tsai, Bing-Heng Yang.

**Writing – original draft:** Li-Tzu Wang, Wu-Chien Chien, Kevin Sheng-Kai Ma, Chi-Hsiang Chung, Yeu-Chin Chen, Wei-Che Tsai, Bing-Heng Yang.

**Writing – review & editing:** Li-Tzu Wang, Wu-Chien Chien, Chi-Hsiang Chung, Bing-Heng Yang.

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
