## [Editor Report · Decision Letter 0]

15 May 2025

Dear Dr Yang,

Thank you for submitting your manuscript entitled "Hydralazine Reduces the Risk of Hematologic Neoplasms in Patients with Hypertension: A Nationwide Population-Based Cohort Study in Taiwan" for consideration by PLOS Medicine.

Your manuscript has now been evaluated by the PLOS Medicine editorial staff. Please note that due to other commitments, the academic editor who provided input on the previous submission has not yet been able to evaluate the responses you have provided. We will wait for their comments before starting the review process. We hope you understand our reasoning.

Please re-submit your manuscript within two working days, i.e. by May 19 2025.

Feel free to email me at atosun@plos.org or us at plosmedicine@plos.org if you have any queries relating to your submission.

Kind regards,

Alexandra Tosun, PhD

Associate Editor

PLOS Medicine

---

## [Decision Letter · Decision Letter 1]

29 Jul 2025

Dear Dr Yang,

Many thanks for submitting your manuscript "Hydralazine Reduces the Risk of Hematologic Neoplasms in Patients with Hypertension: A Nationwide Population-Based Cohort Study in Taiwan" (PMEDICINE-D-25-01670R1) to PLOS Medicine. The paper has been reviewed by subject experts and a statistician; their comments are included below and can also be accessed here: [LINK]

As you will see, the reviewers thought this is a well-conducted study, but also have some concerns. R1 states it does not make sense to match HTN with non-HTN patients, as there was no further analysis of this anywhere in the main paper, and would like an explanation for this. The reviewers also want additional details, and additional discussion of the limitations. After discussing the paper with the editorial team and an academic editor with relevant expertise, I'm pleased to invite you to revise the paper in response to the reviewers' comments. We plan to send the revised paper to some or all of the original reviewers, and we cannot provide any guarantees at this stage regarding publication.

We ask that you submit your revision by Aug 19 2025 11:59PM. However, if this deadline is not feasible, please contact me by email, and we can discuss a suitable alternative.

Don't hesitate to contact me directly with any questions (atosun@plos.org).

Best regards,

Suzanne

Suzanne de Bruijn, PhD

Associate Editor, PLOS Medicine

Sbruijn@plos.org

On behalf of

Alexandra

Alexandra Tosun, PhD

Senior Editor

PLOS Medicine

atosun@plos.org

Comments from the academic editor:

Statistically complex manuscript, which includes both profound strengths (massive sample size, availability of medication data) and frustrating limitations (inability to adjust for BMI. However, I was reassured by the author's response to my earlier comments, which they addressed well.

Comments from the reviewers:

Reviewer #1: This is a well-conducted Nationwide Population-Based Cohort Study in Taiwan on the use of Hydralazine in Reducing the Risk of Hematologic Neoplasms in Patients with Hypertension. The study design, datasets, statistical methods and analyses (especially the competing risk analysis), and presentation (tables and figures) and interpretation of the results are mostly adequate. However, there are still a few issues needing attention.

1. On Page 12, Line 257-260, it says 'We included 1,500,428 non-HTN patients (out of 1,560,379 after 4:1 matching 258 with HTN patients) and 375,107 HTN patients, of whom 59,786 were assigned to the EG and 239,144 (out of 315,321 after 4:1 matching with the EG) were assigned to the RG'. I can see the EG vs RG group matching but it doesn't make sense to match HTN with non-HTN patients, and there was no further analysis anywhere in the main paper regardting HTN vs non-HTN. Can authors please explain the rationale of this HTN vs non-HTN matching?

2. On Page 10, Line 216-220 on multiple-event model. it is unclear exactly how this model was contructed. It was neither mentioned in the statistical analysis section nor presented in the results section in any table or figure. Basically I am not convinced with the multiple-event model approach. It is too complex to deal with simple models. A multi-state makov model could be used in the situation although this could be another paper.

3. On page 9, Line 186-188, it says 'Regarding the exclusion criteria, participants who were aged <20 years, were lost to follow-up, received a diagnosis of HN before the index date, or had missing demographic information were excluded'. However, could author provide a table comparing those included with those excluded to see if there is any bias especialy regarding the loss of follou-ups and missing demographic information?

4. Missing information in the analyses. Behavioural variables such as smoking, drinking and excercise, and formal socioeconomics variables were not included in the analyses, which need to be mentioned in the limitation.

Reviewer #2: Dear Editor,

I would like to thank you for the opportunity that you gave me to review the manuscript "Hydralazine Reduces the Risk of Hematologic Neoplasms in Patients with Hypertension: A Nationwide Population-Based Cohort Study in Taiwan" (PMEDICINE-D-25-01670R1). While the research approach and the findings are quite interesting and well-organized, some issues should be addressed to make it worthy of publication. Please find below my comments and suggestions regarding your work.

Comments

Abstract

-Please also provide the p-values for each comparison that you describe.

Introduction

-Indeed, hypertension constitutes a significant clinical issue in patients with hematological malignancies, especially for allogeneic hematopoietic cell transplantation recipients. Endothelial injury has a critical role in the pathogenesis of hypertension in this setting, and this should be highlighted in the introduction section (https://doi.org/10.1016/j.tru.2024.100186). Moreover, novel agents, such as some TKIs, might lead to the development of drug-induced hypertension (https://doi.org/10.1016/j.blre.2019.03.003).

Methods

-In line 142, I would change "eg" with such as.

-Some more data regarding the definition of health insurance premiums and NTD are essential.

-"Statistical significance was set at a 2-tailed P value of < .001." Why did you use such a p-value and not p<0.05?

Results

-In lines 296-298, please include the exact p-values.

Reviewer #3: This is an interesting paper which used a large national health insurance database in Taiwan and showed that exposure to hydralazine > 180 days was associated with a lower 16-year cumulative incidence of hematologic neoplasms compared to reference group (hydralazine < 180 days).

My comments are as follows:

1. Can you provide dosing information for hydralazine? As a TID medication, patient compliance is often poor with this medication. Did you look into actual patient prescription fill data or assess compliance another way?

2. Hydralazine is not one of the first, second, or third line anti-hypertensive medications. It is used in patients with refractory hypertension or patients with advanced CKD and/or heart failure with reduced ejection fraction. I do not see a comparison of CKD incidence in the EG vs RG. How does your finding translate into clinical practice? Add a brief discussion of the clinical implications.

3. Do you have all-cause mortality data for the study groups? Is there evidence that cancer outcomes are better in the EG?

4. How can you be confident that there is a causative association between long-term hydralazine exposure and reduced incidence of hematologic neoplasms?

---

* Please upload any figures associated with your paper as individual TIF or EPS files with 300dpi resolution at resubmission; please read our figure guidelines for more information on our requirements: http://journals.plos.org/plosmedicine/s/figures. While revising your submission, please upload your figure files to the PACE digital diagnostic tool, https://pacev2.apexcovantage.com/. PACE helps ensure that figures meet PLOS requirements. To use PACE, you must first register as a user. Then, login and navigate to the UPLOAD tab, where you will find detailed instructions on how to use the tool. If you encounter any issues or have any questions when using PACE, please email us at PLOSMedicine@plos.org.

* Please include the URL for each funders website in your Financial disclosure, as well as the initials of the authors who the grants were awarded to.

* Regarding an ethics statement: you state that you don't require an ethics statement, but your study is based on data from human participants. Furthermore, I appreciate the statement in the methods including the IRB approval number and the waiver of informed consent. Please clarify whether you really don't need an ethics statement. Otherwise, include the text 'we've included our ethics statement in the Methods section of our manuscript' in the metadata.

* Please ensure that the study is reported according to the RECORD guideline and include the completed RECORD checklist as Supporting Information. When completing the checklist, please use section and paragraph numbers, rather than page numbers. Please add the following statement, or similar, to the Methods: "This study is reported as per RECORD guideline (S1 Checklist)."

* Author summary: Please include bullet points instead of numbers.

* Authors summary: Please move the author summary to after the abstract in your manuscript.

* Author summary: Please also include limitations in the section 'What Do These Findings Mean?'.

Please see our author guidelines for more information: https://journals.plos.org/plosmedicine/s/revising-your-manuscript#loc-author-summary.

SUPPLEMENTARY MATERIAL

REFERENCES

OBSERVATIONAL STUDIES

* Abstract: Please include the study design, population and setting, number of participants, years during which the study took place (enrollment and follow up), length of follow up, and main outcome measures.

* Please ensure that the study is reported according to the RECORD guideline (available from https://www.record-statement.org) and include the completed checklist as Supporting Information. Please add the following statement, or similar, to the Methods: "This study is reported as per the Reporting of Studies Conducted using Observational Routinely-Collected Data (RECORD) guideline (S1 Checklist)." When completing the checklist, please use section and paragraph numbers, rather than page numbers.

* For all observational studies, in the manuscript text, please indicate: (1) the specific hypotheses you intended to test, (2) the analytical methods by which you planned to test them, (3) the analyses you actually performed, and (4) when reported analyses differ from those that were planned, transparent explanations for differences that affect the reliability of the study's results. If a reported analysis was performed based on an interesting but unanticipated pattern in the data, please be clear that the analysis was data driven.

* Please state in the Methods section whether the study had a prospective protocol or analysis plan. If a prospective analysis plan (from your funding proposal, IRB or other ethics committee submission, study protocol, or other planning document written before analyzing the data) was used in designing the study, please include the relevant document(s) with your revised manuscript as a Supporting Information file to be published alongside your study and cite it in the Methods section. A legend for this file should be included at the end of your manuscript. If no such document exists, please make sure that the Methods section transparently describes when analyses were planned, and when/why any data-driven changes to analyses took place. Changes in the analysis, including those made in response to peer review comments, should be identified as such in the Methods section of the paper, with rationale.

---

## [Decision Letter · Decision Letter 2]

28 Oct 2025

Dear Dr. Yang,

Thank you very much for re-submitting your manuscript "Hydralazine Reduces the Risk of Hematologic Neoplasms in Patients with Hypertension: A Nationwide Population-Based Cohort Study in Taiwan" (PMEDICINE-D-25-01670R2) for review by PLOS Medicine.

Thank you for your detailed response to the reviewers' and editors’ comments. I have discussed the paper with my colleagues, and it has also been seen again by two of the original reviewers. The reviewers were satisfied with the changes made to the paper. However, the editorial team believes that the current reporting does not meet our expectations regarding scientific rigor. Additionally, we believe the study would benefit from a more structured and clear presentation. Please carefully address the editors' comments below in a further revision. When submitting your revised paper, please once again include a detailed point-by-point response to the editorial comments. The remaining issues that need to be addressed are listed at the end of this email.

In revising the manuscript for further consideration here, please ensure you address the specific points made by each reviewer and the editors. In your rebuttal letter you should indicate your response to the reviewers' and editors' comments and the changes you have made in the manuscript. Please submit a clean version of the paper as the main article file. A version with changes marked must also be uploaded as a marked up manuscript file. Please also check the guidelines for revised papers at http://journals.plos.org/plosmedicine/s/revising-your-manuscript for any that apply to your paper.

We ask that you submit your revision within 1 week (Nov 04 2025). However, if this deadline is not feasible, please contact me by email, and we can discuss a suitable alternative.

Please do not hesitate to contact me directly with any questions (atosun@plos.org).

We look forward to receiving the revised manuscript.

Sincerely,

Alexandra Tosun, PhD

Senior Editor 

PLOS Medicine

plosmedicine.org

Comments from Reviewers:

Reviewer #1: Thanks authors for their great effort to improve the manuscript. I am satisfied with the response and revision. No further issues needing attention.

Reviewer #3: All of my comments have been addressed in a satisfactory manner.

Requests from Editors:

GENERAL

* Please confirm that your title complies with to PLOS Medicine's style. Your title must be nondeclarative and not a question. It should begin with main concept if possible. "Effect of" should be used only if causality can be inferred, i.e., for an RCT. Please place the study design ("A randomized controlled trial," "A retrospective study," "A modelling study," etc.) in the subtitle (i.e., after a colon).

* Statistical reporting: Please revise throughout the manuscript, including tables and figures.

- Please report statistical information as follows to improve clarity for the reader ""XX (95% CI [XX,XX]; p</=)"".

- Please separate upper and lower bounds with commas instead of hyphens as the latter can be confused with reporting of negative values.

- Please repeat statistical definitions (HR, CI etc.) for each set of parentheses.

* Please ensure that all abbreviations are defined at first use throughout the text (including statistical abbreviations). Please also check figures and tables.

* Please ensure that tables and figures, including those in supplementary files, are appropriately referenced in the main text.

* Please check that any use of statistical terms (such as trend or significant) are supported by the data, and if not please remove them. Please note that the term trend should be used only when the test for trend has been conducted.

* Please review your text for claims of novelty or primacy (e.g. 'for the first time') and remove this language.

* Your study is observational and therefore causality cannot be inferred. Please remove language that implies causality, such as effect. Refer to associations instead. Please revise throughout.

* Please confirm that all statements (data availability, funding etc.) in the online submission form are accurate.

* Please include the statement on code availability in the data availability statement.

* Please reduce the number of abbreviations throughout the manuscript to improve readability. We suggest to spell out exposure group, reference group, hypertension, hematologic neoplasms, and hydralazine.

* Please revise for use of patient-centered language. Please note that patient-centered language is constructed with the use of post-modified nouns (e.g. 'patients with hypertension’ (or similar) instead of ‘hypertensive patients’) putting the person first in the sentence structure.

* The terms gender and sex are not interchangeable (as discussed in https://www.who.int/health-topics/gender#tab=tab_1 ); please use the appropriate term and revise throughout (including figures and tables).

ABSTRACT

* Please confirm that your abstract complies with our requirements, including providing all the information relevant to this study type https://journals.plos.org/plosmedicine/s/submission-guidelines#loc-abstract

* Please provide the main baseline characteristics of the study population.

* Please confirm that all numbers presented in the abstract are present and identical to numbers presented in the main manuscript text.

* Please reduce the number of abbreviations used in the abstract.

* “adjusted risk difference, RD: 9 [95%CI:2, 16] per 10,000.” – please note that in the following brackets, you have reported the risk difference with a percentage sign.

* “Subgroup analyses revealed lower adjusted sHRs with longer HLZ prescription durations..” – Please specify 'longer duration'.

* “Conversely, users of other anti-HTN medications showed higher adjusted sHRs for overall HN in this cohort.” – we suggest providing one example.

* “dose-dependently” – why dose-dependently?

* Abstract Conclusions:

- Please address the study implications without overreaching what can be concluded from the data; the phrase "In this study, we observed ..." may be useful.

- Please interpret the study based on the results presented in the abstract, emphasizing what is new without overstating your conclusions.

AUTHOR SUMMARY

* Please refer to our previous comment about reducing the number of abbreviations.

* Please streamline the Author Summary. We suggest focusing on the main analysis and removing references to subgroups. The Author Summary should be a nontechnical summary of your research, making your findings accessible to both scientists and non-scientists. For example, "adjusted subdistribution hazard ratio" is not lay language. Also, the Author Summary should be distinct from the scientific abstract.

METHODS AND RESULTS

Editorial note: Throughout the Results section, we found that the findings were not reported with sufficient accuracy and did not always reflect the results presented in the tables. Please note that scientific rigor is of paramount importance to us. Carefully revise the manuscript and consider moving some of the results to the appendix to streamline your manuscript.

* You have stated that statistical significance was set at a 2-tailed P value of <.001. However, you reported significant differences for proteinuria (0.044) and asthma (0.002). Please clarify.

* Table 1: We suggest listing "normal pregnancy" separately from "comorbidities." Also, please define "normal pregnancy”.

* Some of the results that were described as significant (e.g. mebendazole l.305, aspirin l.310) don’t seem to fall within the threshold for statistical significance that you set. Please comment on this and revise accordingly.

* ll.306-309, “Other medications, including itraconazole [44], valproate [45], and metformin [46], which have been reported to inhibit HN progression, exerted a protective effect against HNs, although significance was not reached.” – Based on the results presented in Table 2, we find this statement inaccurate. Please revise.

* l.323ff, “This trend was consistently observed across several HN subgroups, including leukemia of unspecified cell type, lymphoid leukemia, MM and immunoproliferative neoplasms, myeloid leukemia, neoplasm of uncertain 326 behavior, other malignant neoplasms of lymphoid and histiocytic tissue, other polycythemia, and paraproteinemia.” – Please revise for accuracy.

* l.330, “Notably, HLZ demonstrated superior potential in inhibiting the subsequent development of leukemia of unspecified cell type, as shown in the multiple-event model.” – please provide a reference to the relevant table/graph.

* We understand that the paragraph "Association of HN incidence stratified by HN subgroup and duration of HLZ prescription" as well as Table 3 contain a lot of information. We believe that the results could be presented more clearly and with better structure. Please revise accordingly. Consider focusing on a few HN subgroups in the main text and moving some of the results to the supplementary material.

* “HLZ prescription was found to delay HN development” – please note that this statement as currently written implies causality. Please revise.

* “Table 4 presents the results of the sensitivity analysis, which were consistent with those presented in Table 3.” – In the 'first year excluded' analysis, the results for ≥180 days are no longer statistically significant. For the "first five years excluded" analysis, the p-value for ≥180 days is .001, meaning it did not reach statistical significance according to your threshold of <.001.

* “S8 Table revealed…” – the table or the analysis? Please revise for clarity.

* Please provide the unadjusted comparisons as well as the adjusted comparisons in all relevant Tables.

* Please specify the variables controlled for in all relevant Tables (pointing to the relevant section is not sufficient).

DISCUSSION

* Please remove claims of novelty or primacy (e.g. 'for the first time').

* “dose-dependent manner” – Examining different durations is not the same as conducting a dose-response analysis. Please remove these claims throughout the main text.

* “indicating that HLZ may exert favorable effects through several biological regulatory mechanisms.” – How can conclusions about biological regulatory mechanisms be drawn from a retrospective cohort study?

* Please transparently mention whether a study you reference in your discussion was conducted in animals or is based on cell lines.

* l.455-464: Please clarify how this is relevant for your study.

* Due to the retrospective and observational study design, temper any claims regarding clinical use or use in chemoprevention.

General Editorial Requests

---

## [Editor Report · Decision Letter 3]

12 Nov 2025

Dear Dr Yang, 

On behalf of my colleagues and the Academic Editor, Steven C Moore, I am pleased to inform you that we have agreed to publish your manuscript "Association of hydralazine use with risk of hematologic neoplasms in patients with hypertension: a nationwide population-based cohort study in Taiwan" (PMEDICINE-D-25-01670R3) in PLOS Medicine.

I appreciate your thorough responses to the reviewers' and editors' comments throughout the editorial process. We look forward to publishing your manuscript, and editorially there is only one remaining point that should be addressed prior to publication. We will carefully check whether the change has been made. If you have any questions or concerns regarding these final requests, please feel free to contact me at atosun@plos.org.

Please see below the minor point that we request you respond to:

* Table 2: Please check whether ‘Normal pregnancy’ should be written in bold.

Before your manuscript can be formally accepted you will need to complete some formatting changes, which you will receive in a follow up email (including the editorial request above). Please be aware that it may take several days for you to receive this email; during this time no action is required by you. Once you have received these formatting requests, please note that your manuscript will not be scheduled for publication until you have made the required changes.

PRESS

Sincerely, 

Alexandra Tosun, PhD 

Senior Editor 

PLOS Medicine